# Obesity Promotes Renal Inflammation and Fibrosis Independent of Sex in SS Leptin Receptor Mutant (SS^LepR^) Rats

**DOI:** 10.3390/biomedicines13123105

**Published:** 2025-12-17

**Authors:** Karim M. Saad, Mohamed S. Gad, Jocelyn Tang, Kim Capehart, Rafik Abdelsayed, Jan M. Williams, Ahmed A. Elmarakby

**Affiliations:** 1Department of Oral Biology, Dental College of Georgia, Augusta University, Augusta, GA 30912, USA; ksaad@augusta.edu (K.M.S.);; 2Department of Pharmacology and Toxicology, Faculty of Pharmacy, Mansoura University, Mansoura 35516, Egypt; 3Department of Medical Histology and Cell Biology, Faculty of Medicine, Mansoura University, Mansoura 35516, Egypt; 4Department of General Dentistry, Dental College of Georgia, Augusta University, Augusta, GA 30912, USA; kcapehart@augusta.edu; 5Department of Diagnostic Sciences and Pathology, Dental College of Georgia, Augusta University, Augusta, GA 30912, USA; 6Department of Pharmacology and Toxicology, University of Mississippi Medical Center, Jackson, MS 39216, USA

**Keywords:** obesity, renal fibrosis, oxidative stress, inflammation, sex difference, SS^LepR^ mutant rat

## Abstract

**Background**: Obesity is a major contributor to chronic kidney disease (CKD) through mechanisms involving inflammation and metabolic dysregulation. Premenopausal female rats are known to be protected from cardiovascular disorders vs. age matched male rats. The current study investigates if there are sex differences in obesity-induced renal inflammation in SS leptin receptor mutant (SS^LepR^ mutant) rats as a model of metabolic syndrome. **Method**: Male and female lean and obese SS^LepR^ mutant rats were used in the current study to assess changes in metabolic parameters and markers of renal inflammation. **Results**: Obese SS^LepR^ rats showed significant increases in body weight, hemoglobin A1c (HbA1c), and cholesterol vs. lean control, although their blood glucose levels remained comparable to lean rats. Plasma leptin, insulin, and TNF-α converting enzyme (TACE) levels were significantly elevated in obese SS^LepR^ rats vs. lean control rats, with no apparent sex differences. Obesity was associated with an elevation in renal injury since protein and albumin excretion levels were significantly elevated in obese SS^LepR^ rats vs. lean control rats, with no apparent sex differences. The elevation in renal injury was associated with increased renal fibrosis as evidenced by increased collagen deposition and TGF-β expression in the kidney of obese SS^LepR^ rats vs. lean control rats. Increased renal fibrosis also coincided with increased renal inflammation and apoptosis as evidenced by increased macrophage infiltration and IL-6 expression in the kidneys of obese SS^LepR^ rats vs. lean control rats. **Conclusion**: These findings indicate that obesity triggers renal inflammation and fibrosis independent of hyperglycemia in SS^LepR^ rats, and these changes may override sex-based protective effects seen in females in other experimental rodent models of cardiovascular diseases.

## 1. Introduction

Obesity imposes a significant public and economic health burden worldwide, contributing to increased morbidity, mortality, and healthcare expenditures [1,2]. Obesity is a common risk factor for chronic diseases such as type 2 diabetes mellitus [3], hypertension [4], cerebrovascular diseases [5], and cancer [6,7,8]. In addition, obesity increases the risk of musculoskeletal disorders, including osteoarthritis and chronic back pain [9], as well as respiratory complications such as asthma and obstructive sleep apnea [10].

Excessive adipose tissue in obesity promotes insulin resistance, dyslipidemia, and endothelial dysfunction by promoting chronic low-grade inflammation and oxidative stress [11,12]. In particular, visceral adipose tissue can function as an active organ that secretes pro-inflammatory cytokines (e.g., tumor necrosis factor alpha (TNF-α) and interleukin-6 (IL-6)) and adipokines (e.g., leptin and resistin) [13,14]. The dysregulated release of these mediators triggers mitochondrial dysfunction, endoplasmic reticulum stress, and activation of pro-fibrotic signaling pathways, promoting metabolic disturbance and tissue injury [15,16]. Over time, these pathogenic processes culminate in progressive organ dysfunction, manifested by hepatic steatosis, renal impairment, and cardiac remodeling [17].

Recent studies have shown that sex is a key biological variable which influences the onset, severity, and progression of obesity complications [18] and renal injury [19]. Males and females have distinct metabolic and inflammatory profiles due to major differences in body fat distribution and hormonal regulation [20]. However, whether these sex-related differences translate into disparities in obesity-induced renal injury remains incompletely understood, particularly in the context of leptin signaling deficiency.

The Dahl salt-sensitive leptin receptor mutant (SS^LepR^) is advantageous for isolating obesity effects because its genetic setup allows us to bypass variables that often cloud the understanding of obesity’s impact alone, thus offering clearer insights on obesity-induced metabolic and vascular complications. We previously demonstrated that the SS^LepR^ mutant rat displays progressive renal injury as early as 6 weeks of age that was associated with increased macrophage infiltration [21]. Therefore, the current study aims to investigate sex differences in renal inflammation and injury in obese SS^LepR^ mutant rats compared with lean Dahl salt-sensitive controls (SS).

## 2. Materials and Methods

### 2.1. Animal Groups

Six rats per group were used based on a power analysis to detect a 50% difference in renal inflammation between the groups (80% power, α = 0.05). Lean SS and obese SS^LepR^ mutant rats of both sexes (11–12 weeks old) were utilized in the current study. Rats were obtained from Dr. Jan William’s laboratory and were originally developed at the Medical College of Wisconsin using zinc-finger nuclease technology [22].

Rats were housed (3 rats/cage) at the University of Mississippi Medical Center (UMMC) under identical environmental conditions (12 h light/dark cycle, controlled temperature and humidity) with unlimited access to food and water throughout the study. All procedures were conducted under the Animal Welfare Assurance D16-00174 (A3275-01), accredited by the American Association for the Accreditation of Laboratory Animal Care (AAALAC) and UMMC as of 12 July 2024. Genotypes were confirmed by the Genomics Facility at UMMC prior to experimentation. The study involved no treatments, and cages were not rotated; therefore, potential confounders related to treatment or cage location were not specifically controlled. No animals were excluded from the experiment.

Group allocation was inherent to the animals’ genotype (SS and SS^LepR^) and sex; therefore, allocation was known to the investigators from the start. Personnel conducting the experiment and collecting measurements were aware of the group identities, as animals could not be blinded by appearance/genotype. Data analysis was also conducted with the knowledge of group identities.

This study did not involve any procedures expected to cause significant pain or distress; therefore, no specific humane endpoints were established. Animals were monitored daily for general health status, including activity level, posture, coat condition, food and water intake, and signs of illness.

### 2.2. Metabolic Characterization of Female and Male SS and SS^LepR^ Mutant Rats

Twenty-four-hour urine was collected from rats housed individually in metabolic cages. Body weight was recorded before anesthesia with 2% isoflurane, followed by collection of 5–8 mL aortic blood samples. Following blood collection, animals were euthanized by cervical dislocation prior to tissue harvesting, in accordance with institutional guidelines. Blood glucose was measured using a OneTouch Ultra glucometer (LifeScan, Malvern, PA, USA), and HbA1c was determined with the A1cNow Self Check system (PTS Diagnostics, Whitestown, IN, USA).

Plasma was separated and analyzed for cholesterol and insulin using kits from Wako Diagnostics (Richmond, VA, USA) and Mercodia (Winston-Salem, NC, USA), respectively. Leptin (RayBiotech, Comers, GA, USA), TACE (LSBio, Newark, CA, USA), and TNF-α (R&D Systems, Minneapolis, MN, USA) levels were quantified in plasma as indicative of systemic inflammation.

### 2.3. Urinary Biochemical Assays

Urinary protein concentration was measured using a Pierce BCA Protein Assay Kit (Thermo-Fisher, Waltham, MA, USA). Urinary albumin concentration was measured using a commercially available immunoassay kit (Abcam Inc., Waltham, MA, USA). Urinary thiobarbituric acid reactive substances (TBARs) concentration, a marker of renal oxidative stress, was assessed spectrophotometrically according to the manufacturer’s instructions (Cayman Chemical, Ann Arbor, MI, USA). Urinary transforming growth factor beta (TGF-β) concentration was quantified using a TGF-β E-max Immunoassay System (Promega, Madison, WI, USA). All values were then multiplied by 24 h urine volume collected from each animal to determine urinary excretion levels of these parameters per day.

### 2.4. Renal Histopathology

Paraffin kidney sections were prepared from the kidneys collected from SS and SS^LepR^ mutant rats at each time point (3–4 in each group). Kidney sections were cut into 5 mm sections and stained with Masson’s trichrome. To determine the degree of renal fibrosis, 10 representative images per section from each animal were captured at 200× magnification power using a Nikon Eclipse 55i microscope equipped with a Nikon DS-Fi1 color camera (Nikon, Melville, NY, USA). Images were scored blindly on a 1–5 scale to reflect the intensity of blue staining for collagen deposition, as previously described [19].

### 2.5. Immunohistochemistry Studies (IHC)

Paraffin-embedded renal sections were dewaxed, rehydrated, and subjected to antigen retrieval using a sodium citrate solution (pH = 6), before being incubated with hydrogen peroxide (1%) for 30 min. Renal sections were then incubated overnight at 4 °C with primary antibodies for TGF-β (Thermo Fisher, Waltham, MA, USA), IL-6 (Proteintech, Rosemont, IL), F4/80 (Abclonal, Woburn, MA, USA), or caspase-3 (Thermo Fisher, Waltham, MA, USA). After washing with PBS, sections were treated with HRP-conjugated secondary IgG and developed using AEC chromogen (Both from Vector Laboratories, Newark, CA, USA), then counterstained with hematoxylin. For each slide at 200× magnification, 8–10 fields were analyzed. F4/80-positive cells per mm^2^ and percent staining for TGF-β, IL-6, and caspase-3 were quantified using Image J software (Version no. 1.53t).

### 2.6. Caspase-1 Activity Assay

Caspase-1 activity was analyzed with the caspase-Glo 1 inflammasome assay kit (Promega, Madison, WI, USA) according to the manufacturer’s protocol.

### 2.7. Statistical Analysis

Data are expressed as means ± SEM and analyzed using GraphPad Prism Version 10 software (GraphPad Software Inc., La Jolla, CA, USA). Two-way ANOVA was used to evaluate the effect of sex and genotype (obese vs. lean rats) followed by Tukey’s post hoc test. Statistical significance was defined as *p* < 0.05 (* <0.05, ** <0.01, *** <0.001, **** <0.0001). Assumptions for the two-way ANOVA were assessed prior to analysis. Normality of the data was evaluated using the Shapiro–Wilk test. Homogeneity of variances was assessed using standard ANOVA diagnostics. All data met the required assumptions; therefore, no data transformations or alternative statistical methods were necessary.

## 3. Results

### 3.1. Body Weight, Blood Glucose, HbA1c, and Plasma Cholesterol in SS and SS^LepR^ Mutant Rats in Both Sexes

Obese SS^LepR^ rats showed significantly higher body weight, HbA1c, and plasma cholesterol compared with lean SS rats. Bodyweight of obese SS^LepR^ rats in both sexes was significantly higher than their SS counterparts, although females of both strains weighed less than males (Figure 1A). Consistent with prior reports, blood glucose was not significantly different in all rat groups (Figure 1B) [22,23]. On the other hand, HbA1c was significantly elevated only in male SS^LepR^ rats compared with either lean SS males or SS^LepR^ females (Figure 1C), suggesting greater male susceptibility to obesity-related metabolic disturbances. Plasma cholesterol concentrations were significantly higher in SS^LepR^ rats compared with SS rats in both sexes (Figure 1D).

### 3.2. Plasma Leptin, Insulin, TACE, and TNF-α in SS and SS^LepR^ Mutant Rats in Both Sexes

Plasma leptin and insulin concentrations were significantly higher in both male and female obese SS^LepR^ mutant rats compared with their lean SS controls, with no detectable sex differences (Figure 2A,B). Similarly, plasma TACE, the enzyme responsible for converting pro-TNF-α to TNF-α, was elevated in obese SS^LepR^ rats of both sexes relative to their lean SS rat controls (Figure 2C). In contrast, plasma TNF-α levels did not significantly change among the experimental rat groups (Figure 2D).

### 3.3. Total Proteinuria, Albuminuria, and Urinary TBARs in Male and Female SS and SS^LepR^ Mutant Rats

We assess total protein excretion and albuminuria as early markers of renal injury in obesity. Total proteinuria and albuminuria were significantly elevated in obese SS^LepR^ mutant rats compared with lean SS rats with no significant sex difference (Figure 3A,B). Urinary TBARs levels were used as a marker of renal oxidative stress. Obesity increased urinary TBARs excretion levels in both male and female SS^LepR^ mutant rats when compared to their lean control with no apparent sex difference (Figure 3C).

### 3.4. Masson Trichrome Scores and Renal TGF-β Expression in Male and Female SS and SS^LepR^ Mutant Rats

Masson trichrome scoring for collagen deposition (blue staining) was significantly elevated in both male and female obese SS^LepR^ mutant rats compared with lean SS rats with no apparent sex difference (Figure 4A,B). We further confirm the elevation in renal fibrosis in obese SS^LepR^ mutant rats vs. their lean control by assessing renal TGF-β expression and TGF-β excretion levels, a known pro-fibrotic cytokine, in males and females of both rat strains. Consistent with Masson trichrome scoring data, renal TGF-β expression and excretion levels were significantly increased in both male and female obese SS^LepR^ mutant rats compared with lean SS rats with no apparent sex difference (Figure 4C–E). Although male obese SS^LepR^ mutant rats had greater TGF-β than female obese SS^LepR^ mutant rats, these changes were not significant.

### 3.5. Renal F4/80, IL-6 Expression in Male and Female SS and SS^LepR^ Mutant Rats

We used F4/80 immunohistochemical staining as markers of activated macrophages in the kidney sections of lean and obese rats. The number of renal F4/80^+^ cells were significantly elevated in both male and female obese SS^LepR^ mutant rats vs. their lean SS rat counterparts. Male obese SS^LepR^ mutant male rats had a significantly greater number of renal F4/80^+^ cells than female obese SS^LepR^ mutant rats (Figure 5A,B). Immunohistochemical assessment of the pro-inflammatory cytokine IL-6 also revealed a significant increase in renal IL-6 expression in both SS^LepR^ mutant rat sexes relative to their lean SS controls with no apparent sex difference (Figure 5C,D).

### 3.6. Renal Caspase-1 Activity and Caspase-3 Expression in SS and SSLepR Mutant Rats in Both Sexes

Caspase-1 activation triggers pyroptosis, an inflammatory form of programmed cell death that contributes to the progression of renal injury [24]. In the current study, renal caspase-1 levels were significantly higher in obese SS^LepR^ mutant rats of both sexes compared with their lean SS counterparts, with no detectable sex differences (Figure 6A). Obese SS^LepR^ rats of both sexes showed a significantly higher expression of central executioner caspase-3 compared to lean SS rats, without a sex-specific difference.

## 4. Discussion

Obesity is a major public health concern that contributes to the development of renal injury and the progression of CKD through diverse mechanisms involving elevation in oxidative stress, inflammation, and metabolic dysregulation. Previous studies have shown that obesity promotes glomerular hypertrophy, proteinuria, and renal fibrosis, which eventually lead to renal dysfunction and end-stage kidney disease [25,26,27,28]. In the current study, we utilized the obese SS^LepR^ mutant rat, a leptin receptor-deficient model on the Dahl salt-sensitive (SS) genetic background, to investigate obesity-induced renal injury and to assess potential sex differences independent of confounding factors such as hyperglycemia and hypertension. Our findings demonstrate that obese SS^LepR^ mutant rats exhibit significant elevations in body weight, HbA1c, cholesterol, leptin, and insulin vs. their lean SS counterparts, which confirms the metabolic dysregulation characteristic of this model [22,29]. Sex difference was only observed in body weight, HbA1c, and renal macrophage infiltration where obese male SS^LepR^ mutant rats had greater body weight, HbA1c, and renal macrophage infiltration than obese females. Furthermore, we observed significant increases in renal injury markers, inflammation, and fibrosis in obese SS^LepR^ mutant rats vs. their lean SS counterparts, with no apparent sex differences. Our findings suggest that obesity alone is sufficient to initiate significant renal histopathological changes and injury, at least, in our obese SS^LepR^ mutant rat model. Our findings also suggest that renal vascular protection observed in females vs. males in many rodent models of cardiovascular diseases is lost in obese female SS^LepR^ mutant rats.

The obese SS^LepR^ mutant rat model was created on the Dahl SS genetic background using zinc-finger technology. The obese SS^LepR^ mutant rat model develops hyperlipidemia and hyperinsulinemia by 6 weeks of age, without hyperglycemia, similar to clinical features of metabolic syndrome [21,22,23]. Based on our previous findings, these rats reach sex maturity at the age of 10 weeks [22]. Thus, in the current study, 11–12-week-old male and female obese SS^LepR^ mutant and lean SS rats were utilized to address sex differences in metabolic parameters and renal injury independent of hyperglycemia and hypertension. Consistent with the previously published data [22,29], body weight, cholesterol, and insulin levels were elevated in the obese SS^LepR^ mutant strain relative to the lean SS rat strain. Sex differences in body weight and HbA1c were observed in obese SS^LepR^ mutant rats, whereby males had higher body weight and HbA1c than females, suggesting that obese male SS^LepR^ mutant rats might be more sensitive to obesity-induced metabolic dysregulation than females. Since both male and female SS^LepR^ mutant rats are hyperinsulinemic but not hyperglycemic or hypertensive at 11–12 weeks old [22,29], this model is ideal to address the effect of obesity on renal injury independent of hyperglycemia or hypertension.

Leptin is a hormone produced by fat cells to regulate body weight and appetite, and studies have shown that higher leptin levels are associated with insulin resistance and elevation in body mass index (BMI) [30]. The role of leptin in the regulation of body weight and appetite is well established. Clinical and experimental findings have shown that higher levels of leptin are associated with insulin resistance and increased BMI [30]. Consistent with previous findings, plasma levels of leptin and insulin were significantly elevated in both male and female obese SS^LepR^ mutant rats vs. lean SS rats; however, there were no apparent sex differences. TNF-α is a key cytokine released from adipocytes and is known to dysregulate leptin and inhibit glucose-stimulated insulin secretion leading to insulin resistance [31]. TACE is the enzyme that cleaves the membrane-bound precursor protein pro-TNF-α to release TNF-α [25]. Plasma TACE level is known to be increased in obese subjects, particularly those associated with insulin resistance [32]. Although plasma TACE levels were elevated in male and female obese SS^LepR^ mutant rats vs. lean SS rats, there was no difference in plasma TNFα levels. Since TNF-α is mainly released by adipocytes, it could increase locally in obesity rather than being systemically elevated in the plasma. Additionally, TNF-α has a short half-life in circulation as it can rapidly be cleared by the soluble TNF-α receptor [33]. Thus, the plasma TNF-α level is considered an unreliable marker of tissue inflammation in obesity.

Previous findings have shown that the obese SS^LepR^ mutant rat developed progressive proteinuria and renal injury at 8 weeks of age relative to the lean SS rats, and males tended to have a higher degree of renal injury than females [22]. Elevation in oxidative stress is known to contribute to obesity-induced renal injury [34]. In the current study, we assessed proteinuria and albuminuria as markers of renal injury as well as TBARs excretion as a marker of renal oxidative stress. As shown previously [22], obese SS^LepR^ mutant rats had a significant elevation in proteinuria and albuminuria vs. lean SS rats, with no sex differences in these parameters, although our rats were 4 weeks older than this study. Renal TBARs excretion was also significantly elevated in obese SS^LepR^ mutant rats vs. lean SS rats, with no apparent sex differences

It is well known that renal fibrosis is a progressive process that leads to the loss of kidney function and end-stage renal disease [35]. In line with the previous literature [22,36], obese SS^LepR^ mutant rats in the current study exhibited significant renal structural injury, as evidenced by increased Masson trichrome scoring for collagen deposition. The upregulation of renal TGF-β expression, a master regulator of fibrosis, was also observed in obese SS^LepR^ mutant rats vs. lean SS rats, and this elevation coincided with increased urinary TGF-β excretion in our obese rat model. TGF-β is known to promote ECM accumulation, mesangial expansion, and glomerulosclerosis, which are hallmarks of obesity-related nephropathy [37,38,39]. The observed increase in TGF-β expression in SS^LepR^ mutant rats is consistent with findings in other models of metabolic syndrome-associated renal injury and reinforces the hypothesis that chronic inflammatory and oxidative stress activates profibrotic signaling pathways in the kidney during cardiovascular diseases [32,40,41].

Obesity contributes to endothelial dysfunction and renal damage by activating pro-immune cells infiltration and inflammatory pathways [42,43,44,45]. Infiltration of monocytes and their differentiation to macrophages is known to be augmented in obesity [46]. Macrophages play an important role in the pathogenesis and progression of renal injury and fibrosis via increased oxidative stress, growth factors, and pro-inflammatory cytokines [22]. We have previously demonstrated that the progressive proteinuria in both male and female obese SS^LepR^ mutant rats coincided with increased renal macrophage infiltration as early as 6 weeks of age [22]. Macrophage depletion slows the progression of proteinuria and renal injury in obese SS^LepR^ mutant rats [21]. Consistent with these findings, renal macrophage infiltrations were significantly elevated in obese SS^LepR^ mutant rats vs. their lean SS rat counterparts, and obese males had significantly greater macrophage infiltration than females. IL-6 is involved in renal inflammation and injury in cardiovascular disease including obesity, and macrophages can produce IL-6 [47]. Renal IL-6 expression levels were also elevated in obese SS^LepR^ mutant rats vs. their lean SS rat counterparts, with no apparent sex difference, suggesting local inflammatory signal activation in the kidney. Caspase-1 is a key inflammatory enzyme in the activation of inflammasomes, which are multi-protein complexes that respond to cellular damage and stress in kidney diseases [24]. Caspase-3 is a key executioner enzyme of apoptosis (programmed cell death) that contributes to tissue damage in both acute kidney injury (AKI) and chronic kidney disease (CKD) [48]. The upregulation of both renal active caspase-1 and renal caspase-3 expression in obese SS^LepR^ mutant rats vs. their lean SS rat counterparts further supports an interplay between inflammation and cell death. Since caspase-1 is central to inflammasome activation and pyroptosis [49] whereas caspase-3 mediates apoptotic pathways [50], the concurrent activation of these pathways suggests that both pyroptotic and apoptotic mechanisms contribute to renal injury in obesity, which ultimately exacerbates fibrosis and tissue remodeling.

Our study has several limitations. First, the hormone levels or estrous cycle stages in females were not assessed, which could influence inflammatory responses. Second, our study only assessed changes in the renal expression of macrophages without assessing other immune cells, such as T cells, which are known to play a crucial role in renal inflammation and injury. Finally, we have not targeted oxidative stress, inflammation, and/or fibrosis as key pathways for the progression of renal injury in our obese animal model. Thus, future studies will examine if pharmacological inhibition of key inflammatory and fibrotic signals such as TGF-β and IL-6 and/or apoptotic pathways, such as caspase-1 and caspase-3 inhibitors, will slow the progression of proteinuria and renal injury in our obese SS^LepR^ mutant rat model. Understanding the role of these signals could uncover novel therapeutic strategies to mitigate obesity-related renal injury through modulation of TGF-β and/or apoptotic pathways.

In conclusion, our findings demonstrate that obesity in *SS*^LepR^ mutant rats induces systemic metabolic dysregulation and increased renal inflammation and fibrosis leading to significant degrees of renal injury. These effects occurred independently of hyperglycemia and hypertension, confirming that obesity can directly induce renal histopathological changes that lead to renal injury. Although sex differences were evident in body weight, HbA1c, and renal macrophage infiltration, there were no apparent sex differences in overall renal inflammation and injury in our obese SS^LepR^ mutant obese rats model, which is in contrast with clinical and experimental studies that suggest greater susceptibility of males vs. females to obesity-induced renal vascular injury [37,51]. We postulate that the absence of the leptin receptor in the *SS*^LepR^ mutant rats model produces a level of metabolic and inflammatory stress that overrides sex-based protective mechanisms seen in females vs. males in other cardiovascular disease models.

## Figures and Tables

**Figure 1 biomedicines-13-03105-f001:**
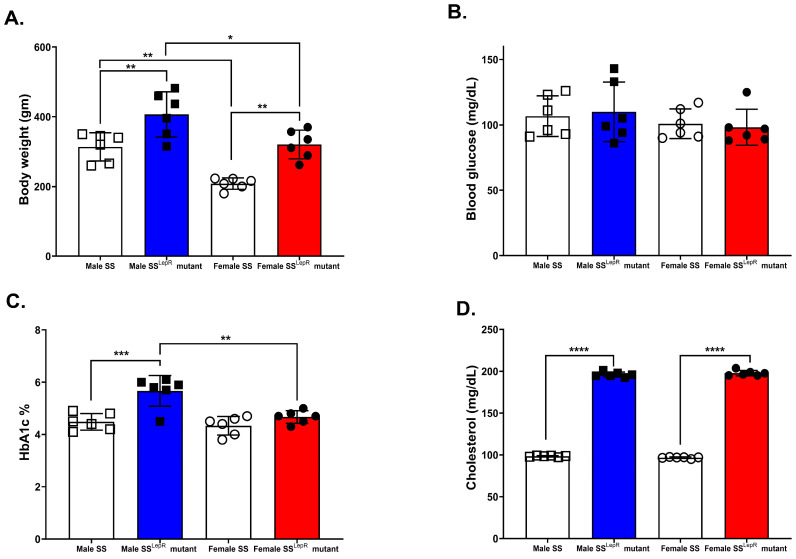
Sex-specific differences in obesity-associated increases in body weight and HbA1c. Metabolic measurements in male and female lean Dahl salt-sensitive (SS) and obese SS leptin receptor mutant (SS^LepR^ mutant) rats at 11–12 weeks of age. (**A**) Body weight; (**B**) Blood glucose; (**C**) HBA1c%; (**D**) Plasma cholesterol. Data are presented as means ± SEM. * *p* < 0.05, ** *p* <0.01, *** *p* < 0.001, **** *p* < 0.0001 (*n* = 6/group).

**Figure 2 biomedicines-13-03105-f002:**
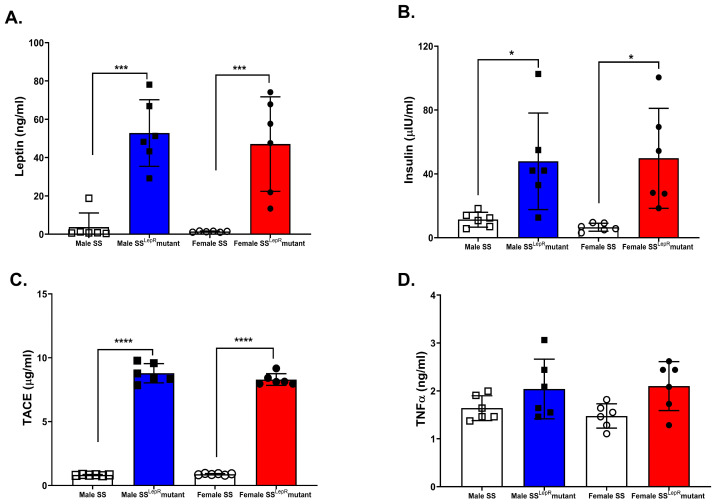
Obesity-induced increase in systemic markers of inflammation and insulin resistance in both sexes. Plasma concentrations of leptin (**A**), insulin (**B**), TACE (**C**), and TNF-α (**D**) in male and female lean Dahl salt-sensitive (SS) and obese SS leptin receptor mutant (SS^LepR^ mutant) rats. Data are expressed as mean ± SEM. * *p* < 0.05, *** *p* < 0.001, **** *p* < 0.0001 (*n* = 6/group).

**Figure 3 biomedicines-13-03105-f003:**
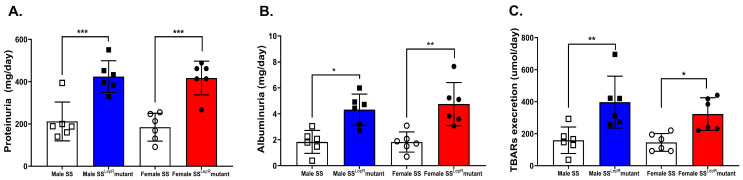
Obesity increases indices of renal injury and oxidative stress in both sexes. Protein excretion (**A**) and albuminuria (**B**) were assessed as markers of renal injury, and TBARs excretion levels (**C**) were also assessed as a marker of renal oxidative stress in male and female lean Dahl salt-sensitive (SS) and obese SS leptin receptor mutant (SS^LepR^ mutant) rats. Values are presented as means ± SEM. * *p* < 0.05, ** *p* < 0.01, *** *p* < 0.001 (*n* = 6/group).

**Figure 4 biomedicines-13-03105-f004:**
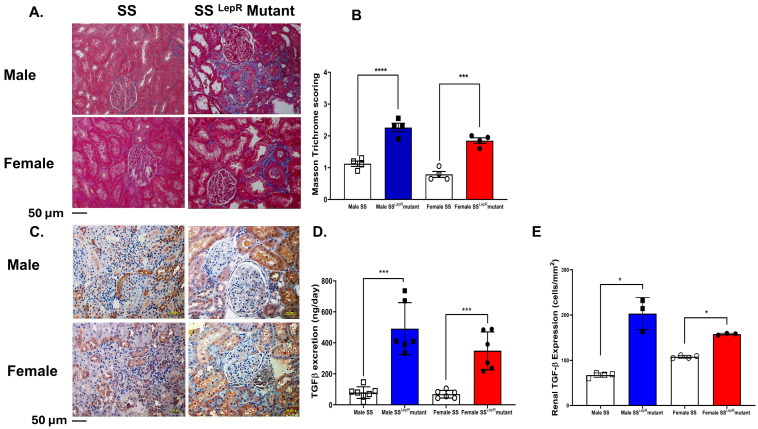
Obesity increases renal fibrosis and TGF-β expression in both sexes with no apparent sex difference. (**A**) Representative images for Masson trichrome staining of kidney sections from both male and female lean Dahl salt-sensitive (SS) and obese SS leptin receptor mutant (SS^LepR^ mutant) rats captured at 200× magnification and average of Masson trichrome scoring for intensity of collagen deposition (**B**). Representative images of TGF-β expression in kidney sections at 200× magnification (**C**) and area % of TGF-β expression (**D**) in both male and female lean SS and obese SS^LepR^ mutant rats (*n* = 3–4/group). (**E**) Urinary TGF-β excretion levels stress in male and female lean SS and obese SS^LepR^ mutant rats. Values are presented as mean ± SEM (*n* = 6/group, * *p* < 0.05, *** *p* < 0.001, **** *p* < 0.000).

**Figure 5 biomedicines-13-03105-f005:**
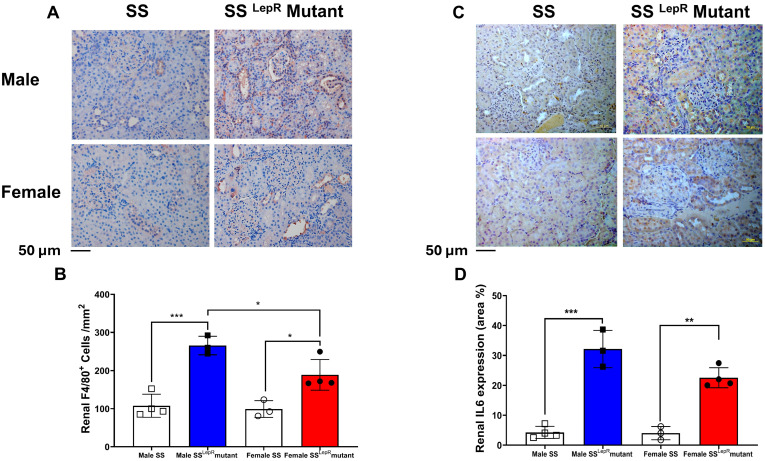
Obesity-induced increase in renal macrophages infiltration and IL-6 expression in both sexes. Representative images of immunohistochemical staining of F4/80+ cells at 200× magnification (**A**) and average number of F4/80 positive cells per mm2 (**B**) as indicative of macrophages infiltration in both male and female lean Dahl salt-sensitive (SS) and obese SS leptin receptor mutant (SS^LepR^ mutant) rats. Representative images of immunohistochemical staining of renal IL-6 expression at 200× magnification(**C**) and area percentage of renal IL-6 expression in male and female lean SS and obese SS^LepR^ mutant rats (**D**). Values are presented as mean ± SEM (*n* = 3–4/group, * *p* < 0.05, ** *p* < 0.01, *** *p* < 0.001).

**Figure 6 biomedicines-13-03105-f006:**
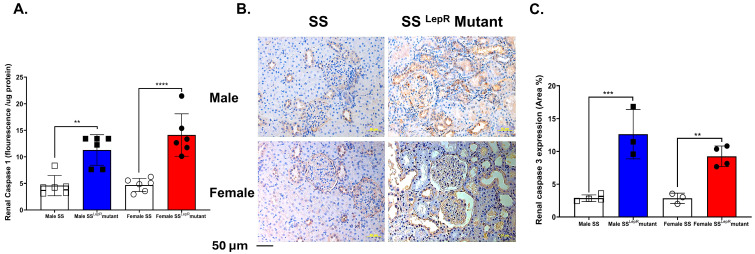
Obesity increases markers of renal inflammation and apoptosis in both sexes. Renal caspase-1 fluorescence intensity relative to protein concentration (**A**) in male and female lean Dahl salt-sensitive (SS) and obese SS leptin receptor mutant (SS^LepR^ mutant) rats (*n* = 6/group). Representative images of immunohistochemical staining of renal caspase-3 at 200× magnification (**B**) and area percentage of caspase-3 expression (**C**) in both male and female lean SS and obese SS^LepR^ mutant rats (*n* = 3–4/group). Values are presented as mean ± SEM. (*n* = 3–4/group, ** *p* <0.01, *** *p* < 0.001, **** *p* < 0.000).

## Data Availability

Any data will be made available upon request.

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
