# Peer review of "Obesity Promotes Renal Inflammation and Fibrosis Independent of Sex in SS Leptin Receptor Mutant (SSLepR) Rats"

_biomedicines, 2025, doi:10.3390/biomedicines13123105_

Round 1

Reviewer 1 Report

Comments and Suggestions for Authors

This study aims to demonstrate that obesity itself is sufficient to induce kidney inflammation, fibrosis, and injury. However, there are some significant issues with the completeness of experimental evidence and the depth of interpretation of some data need to be addressed.

1. The authors evaluated the impact of obesity on renal injury (such as kidney damage, renal inflammation, and fibrosis) from multiple perspectives. However, each section remains superficial, and the indicators examined are insufficient. For instance, during the assessment of renal function, serum urea nitrogen and creatinine levels were not measured; in the case of fibrosis, only TGF-β was detected, without evaluating collagen expression (such as collagen I and collagen IV); and for apoptosis, only caspase 1 and caspase 3 were briefly examined. Please explain the reasons for these omissions.

2. Could the renal function impairment, inflammation, and fibrosis in obese SS leptin receptor mutant (SSLepR mutant) rats be attributed to the genetic mutation itself, rather than solely to obesity?

3. Only urine protein, urine albumin, and urine TBARs were used as markers of kidney injury in the article. Lack of the most directly and clinically relevant blood indicators for evaluating glomerular filtration function - serum creatinine and blood urea nitrogen. This makes the assessment of the degree of 'kidney injury' incomplete.

4. Why was H&E staining of renal tissues not performed (a key indicator for evaluating renal injury)?

5. It is recommended to use TUNEL staining to detect apoptosis in renal tissue cells.

6. In Figure 2, TACE is significantly increased, so why is there no change in TNF-α?

7. In Figure 4A and 4C, the magnification of the images is too high, resulting in a limited field of view. It is recommended to supplement with images at lower magnifications.

8. In Figure 5B, the macrophage infiltration (F4/80+cells) in male obese rats was significantly higher than that in females. The author only mentions in the article, but still insists on the overall conclusion that there is no significant difference. This may cause controversy.

9. Some of the journal name abbreviations in the references are not standardized, with some using full names and others using abbreviations.

Author Response

We would like to thank the reviewer for his valuable comments. We have revised the manuscript based on these comments and below is our response to the comments

1- Our study is a follow up to previous findings of Dr. William's laboratory, where they previously reported increased in creatinine clearance as indicative of kidney function in our obese rats with no apparent sex differences. So, we have not repeated it in this study.

2- Since we only noticed significant changes in serum urea nitrogen in models of acute renal injury such as IR model, we did not run this assay as we do not expect changes BUN in chronic model of renal inflammation.

3- Similarly, we have not stained for H & E since Dr. Williams already published this before. we used Masson trichrome as indication for collagen deposition and both TGF-beta expression and excretion for ECM deposition as indicative of renal fibrosis. however, we run collage 4 IHC and data were similar to Masson trichrome and TGF beta (collagen 4 increased in obesity with no apparent sex differences and we can add that as data supplement 

4- We have not run TUNEL assay in our study since we did not prepare any frozen kidney sections. Paraffin embedded sections in rats did not give good results based on our previous expertise with this assay. Thus, we believe that caspase 1 and caspase 3 are good indication for changes in renal apoptosis and pyroptosis. 

5-The reviewer raised excellent point. we do not think that inflammation, fibrosis and injury in obese SS leptin receptor mutant (SSLepR mutant) rats could be attributed to the genetic mutation itself since we use Dahl SS rats as control with same genetic background. Thus, we believe that changes in renal injury in our model are mainly attributed to obesity. 

6- Since TNF-α is mainly released by adipocyte, it could increase locally in obesity rather than systemically elevated in the plasma. Additionally, TNF-α has short half-life in circulation as it can rapidly be cleared by the soluble TNF-α receptor. Thus, changes in plasma TNF-α level might not be noted in obesity.

7- We also used 200X magnification in our previous studies especially when counting Macrophages as well as when assessing % area of expression to make it clear to readers in publication. However, intensity of staining at 100X was not strong enough since studies were done last year. We could provide more images at 200 X as supplement file if needed. 

8. Figure 5B, we totally agreed with the reviewer that the macrophage infiltration (F4/80+cells) in male obese rats was significantly higher than that in females indicating sex difference in renal macrophages.  We apologize for the confusion, and we have revised this in the discussion of the manuscript. 

9- We thank reviewer for his valuable insights. We used Endnote to create library and insert references based on MDPI journal style. If the journal requests any changes, we will be glad to revise references manually. 

Reviewer 2 Report

Comments and Suggestions for Authors
  1. When an abbreviation appears for the first time, the full name should be indicated. Please check carefully. For example, "HbA1c" in page1 line 28. Moreover, the full names and abbreviations are used interchangeably throughout the text, such as "hemoglobin A1c" and "HbA1c".
  2. There are issues such as grammar. Please check carefully. For example, page 2 line 49-50 should be "Obesity is a common risk factor..."
  3. The "Statistical Analysis" section is relatively brief. It is suggested that it be described more clearly.
  4. Why are there six rats in each group? What is the basis for this calculation?

Author Response

We would like to thank the reviewer for his valuable comments. We have revised the manuscript accordingly and all revision are now highlighted in yellow. 

1- We have checked all the abbreviation and make sure full name is spelled out when it appeared for first time. 

2- We have run grammar check and fix errors. 

3- We have revised statistical section based on the reviewer comments. 

4- A power analysis was performed, and it was estimated that n=6 rats/group would have 80% power (α=0.05) to detect a 50% difference in renal inflammation between groups

Reviewer 3 Report

Comments and Suggestions for Authors

This study continues previously published data on structural renal injury in the context of obesity in an experimental model (obese SSLepR mutant rats vs. lean SS rats). The manuscript is generally well-structured and is associated with robust statistical analyses, providing significant data in this area of ​​research – the impact of obesity on renal function.

However, to increase the quality of the manuscript, I have the following suggestions for the authors:

- Please reconsider shortening the Abstract to around 250 words, in accordance with Biomedicines journal's recommendations.

- Please provide the Ethical Approved Numbers for this study in Materials and Methods, Animal groups subsection.

- Please provide the mL of blood samples collected from the aorta to each animal in the Materials and Methods section.

- Please expand the Urinary Biochemical Assays section with more details regarding all urinary proteins analysed.

- Lines 174-177 are more suitable for the Discussion section. The same suggestion is for lines 193, 206-207, 249-253, and 256-258

- To be more conclusive and comparative for readers, please provide comparative images for TGF immunoexpression (all including or not renal corpuscles in Figure 4C. Also, please increase their typographic quality (especially 4C – Female SS and SSLepR Mutant)

- It is difficult to identify the F4/80+ cells in Figure 5, so please improve the typography of these figures. Please also explain the moderate positive F4/80+ reaction in the tubular epithelial cells in the same Figure.

- Kindly also increase the typographical quality of Figure 5B (IL-6 immunoepression in kidney)

- Please add the limits of this study at the end of the Discussion Section and organise a Conclusions section in the main text of the manuscript.

Author Response

We would like to thank the reviewer for his valuable suggestion. We have revised the manuscript accordingly and all changes are now highlighted in yellow 

1- Abstract has been shortened based on the reviewer suggestion and journal guideline

2- Ethical Approved Numbers has been added to method section 

3- 5-8 mL of blood were collected from the aorta from each animal and plasma was separated from this blood.

4- Urinary Biochemical Assays section was expanded based on reviewer suggestion. 

5- We have revised results section and moved it to discussion based on reviewer comments 

6- Comparative images for TGF immuno-expression with improved typographic quality are now provided

7- Comparative images for F4/80 immuno-expression with improved typographic quality are now provided. we also agreed with reviewer that positive F4/80+ was noted in not only in proximal tubular but also in distal tubular cells.

8- We have increased typographical quality of IL-6 immuno-expression in kidney based on the reviewer's comment. 

9- We have added the limits of this study at the end of the Discussion Section and organized a conclusions section in the main text of the manuscript.

Round 2

Reviewer 1 Report

Comments and Suggestions for Authors

The author has already answered my questions. 

Author Response

We would like to thank the reviewer for the effort he/she puts to review our work 

Reviewer 3 Report

Comments and Suggestions for Authors

The authors provide an improved version of the manuscript, which, in my opinion, can be accepted for publication in Biomedicines only if the quality of the microscopic figures allows it (according to the journal rules). I cannot appreciate this, as they appear at a reduced resolution and size.

Author Response

All images were provided as 300x300 dpi as per the journal instructions. However, the PDF provided for reviewer had reduced image quality.  we have attached a presentation for the representative images with high quality to address reviewer concerns and we will make it available as well for journal to use in the manuscript. 
